# Pharmacological Inhibition of WIP1 Sensitizes Acute Myeloid Leukemia Cells to the MDM2 Inhibitor Nutlin-3a

**DOI:** 10.3390/biomedicines9040388

**Published:** 2021-04-06

**Authors:** Maria Chiara Fontana, Jacopo Nanni, Andrea Ghelli Luserna di Rorà, Elisabetta Petracci, Antonella Padella, Martina Ghetti, Anna Ferrari, Giovanni Marconi, Simona Soverini, Ilaria Iacobucci, Cristina Papayannidis, Antonio Curti, Ernesta Audisio, Maria Benedetta Giannini, Michela Rondoni, Francesco Lanza, Michele Cavo, Giovanni Martinelli, Giorgia Simonetti

**Affiliations:** 1IRCCS Istituto Romagnolo per lo Studio dei Tumori “Dino Amadori”—IRST, 47014 Meldola (FC), Italy; elisabetta.petracci@irst.emr.it (E.P.); antonella.padella@irst.emr.it (A.P.); martina.ghetti@irst.emr.it (M.G.); anna.ferrari@irst.emr.it (A.F.); giovanni.marconi@irst.emr.it (G.M.); maria.giannini@irst.emr.it (M.B.G.); giovanni.martinelli@irst.emr.it (G.M.); giorgia.simonetti@irst.emr.it (G.S.); 2Dipartimento di Medicina Specialistica, Diagnostica e Sperimentale, Università di Bologna, 40138 Bologna, Italy; jacopo.nanni2@studio.unibo.it (J.N.); simona.soverini@unibo.it (S.S.); michele.cavo@unibo.it (M.C.); 3IRCCS Azienda Ospedaliero-Universitaria di Bologna, Istituto di Ematologia “Seràgnoli”, 40138 Bologna, Italy; cristina.papayannidis@unibo.it (C.P.); antonio.curti2@unibo.it (A.C.); 4Department of Pathology, St. Jude Children’s Research Hospital, Memphis, TN 38105, USA; ilaria.iacobucci2@unibo.it; 5AOU Città della Salute e della Scienza di Torino, 10126 Torino, Italy; eaudisio@cittadellasalute.to.it; 6Hematology Unit & Romagna Transplant Network, Ravenna Hospital, 48121 Ravenna, Italy; michela.rondoni@auslromagna.it (M.R.); francesco.lanza@auslromagna.it (F.L.)

**Keywords:** AML, novel therapeutic targets, *WIP1*, *MDM2*

## Abstract

In acute myeloid leukemia (AML), the restoration of p53 activity through MDM2 inhibition proved efficacy in combinatorial therapies. WIP1, encoded from *PPM1D*, is a negative regulator of p53. We evaluated *PPM1D* expression and explored the therapeutic efficacy of WIP1 inhibitor (WIP1i) GSK2830371, in association with the MDM2 inhibitor Nutlin-3a (Nut-3a) in AML cell lines and primary samples. *PPM1D* transcript levels were higher in young patients compared with older ones and in core-binding-factor AML compared with other cytogenetic subgroups. In contrast, its expression was reduced in *NPM1*-mutated (mut, irrespective of *FLT3*-ITD status) or *TP53*-mut cases compared with wild-type (wt) ones. Either Nut-3a, and moderately WIP1i, as single agent decreased cell viability of *TP53*-wt cells (MV-4-11, MOLM-13, OCI-AML3) in a time/dosage-dependent manner, but not of *TP53*-mut cells (HEL, KASUMI-1, NOMO-1). The drug combination synergistically reduced viability and induced apoptosis in *TP53*-wt AML cell line and primary cells, but not in *TP53*-mut cells. Gene expression and immunoblotting analyses showed increased p53, MDM2 and p21 levels in treated *TP53*-wt cells and highlighted the enrichment of MYC, PI3K-AKT-mTOR and inflammation-related signatures upon WIP1i, Nut-3a and their combination, respectively, in the MV-4-11 *TP53*-wt model. This study demonstrated that WIP1 is a promising therapeutic target to enhance Nut-3a efficacy in *TP53*-wt AML.

## 1. Introduction

Protein Phosphatase, Mg2^+^/Mn2^+^ Dependent 1D (*PPM1D*) is a member of the PP2C family of serine/threonine phosphatase and encodes for “Wild-Type p53-Induced Phosphatase 1” (WIP1). WIP1 is involved in the negative regulation of stress response pathways [1,2], DNA damage response (DDR) [3,4,5] and cell-cycle [6,7,8] Following DNA damages, p53 is activated and promotes the transcription of several downstream DDR-effectors including *PPM1D*. *PPM1D* dephosphorylates p53 at Ser15 [9], thus promoting the interaction with its negative regulators MDM2 and MDMX [10,11,12]. This autoregulatory feedback loop allows the termination of p53 response after DNA damages [11,13].

*PPM1D* amplification and WIP1 overexpression showed oncogenic properties across several cancer types and were associated with dismal outcome, mainly due to the suppression of p53 activity [14,15,16,17,18,19,20,21,22].

Moreover, *PPM1D* gain-of-function mutations are enriched in peripheral blood cells of individuals that have been previously exposed to chemotherapy [23,24], in therapy-related myelodysplastic syndromes and in clonal hematopoiesis of indeterminate potential [25,26,27]. *PPM1D* mutations confer advantages to hematopoietic stem cells in terms of self-renewal and/or proliferation, resulting in the expansion of multi-lineage and myeloid-based clones [25,26,27,28]. It has been recently reported that truncating *PPM1D* mutations that induce elevated protein expression, confer cytarabine resistance in AML and force the selective expansion of *PPM1D*-mutated leukemic cells [29]. Allosteric WIP1 inhibition can restore the sensitivity of *PPM1D*-mutated leukemia to chemotherapy. These preliminary data provide the rationale for exploiting the beneficial effect of WIP1 inhibitors in combinatorial therapies. The WIP1 inhibitor GSK2830371 (WIP1i) has been widely tested as single agent or in combination with chemotherapeutic or targeted drugs in pre-clinical studies on different neoplastic cell-lines, showing promising results [30,31,32,33]. Burdova et al. have recently demonstrated that WIP1 inhibition induces an accumulation of DNA damage in S/G2 cells and sensitizes cancer cells to olaparib [32], a poly ADP ribose polymerase inhibitor.

In leukemic cells, genomic instability is frequently linked to structural or functional p53 abnormalities [34,35,36]. Several mechanisms underlying non-mutational p53 inactivation might carry therapeutic relevance. The restoration of its activity through inhibition of the E3 ubiquitin-protein ligase binding to p53 (MDM2) has been extensively studied in the past years, as pharmacological intervention against AML that retains wild-type (wt) p53 [37,38,39]. Moreover, enhanced cytotoxicity and apoptotic response were observed by combining MDM2 inhibitors with chemotherapeutic (e.g., cytosine arabinoside and doxorubicin) or targeted agents (e.g., *FLT3*, *MEK1* or *BCL*2 inhibitors) [40,41] and phase I/II studies are ongoing (NCT02670044; NCT03850535; NCT04029688). Combinatorial inhibition of MDM2 and WIP1 enhanced tumor growth-inhibitory and cytotoxic activity of MDM2 inhibitors in melanoma, neuroblastoma and breast cancer [30,32,33]. However, their combined activity in leukemia cells has not been investigated yet.

Thus, we here assessed the use of WIP1i in enhancing the therapeutic response of AML cell lines and primary cells to the MDM2 inhibitor Nutlin-3a and we elucidated the molecular mechanism underlying its action.

## 2. Experimental Section

### 2.1. Human AML Cell Lines

OCI-AML3 (*DNMT3A*-mut, *NPM1*-mut, *TP53*-wt), MOLM-13 (*KMT2A*-rearranged, *TP53*-wt), MV-4-11 (*KMT2A*-rearranged, *FLT3*-ITD, *TP53*-wt), NOMO-1 (*KMT2A*-rearranged, *TP53*-mut), HEL (*TP53*-mut, *JAK2*-mut) and KASUMI-1 (*RUNX1*-*RUNXT1*, *TP53*-mut) cell lines were obtained from Leibniz-Institut DSMZ-Deutsche Sammlung von Mikroorganismen und Zellkulturen GmbH (Braunschweig, Germany) and American Type Culture Collection (ATCC, Gaithersburg, MD, USA), respectively, and cultured following manufacturer’s recommendations. MOLM-13, MV-4-11, HEL, KASUMI-1 and NOMO-1 were cultured in RPMI-1640 medium (Euroclone, Milano, Italy), while OCI-AML3 were cultured in alpha-MEM (Thermo Fisher Scientific, Whaltam, MA, USA) in a humidified atmosphere of 5% CO2 at 37 °C. Media were supplemented with 2 mM L-glutamine (GE Healthcare, Chicago, IL, USA), 10–20% heat-inactivated fetal bovine serum (Thermo Fisher Scientific, Whaltam, MA, USA), 100 U/mL penicillin, 100 μg/mL streptomycin (GE Healthcare, Chicago, IL, USA).

### 2.2. Human Primary Cells

Primary AML blast cells were obtained upon written informed consent, as approved by the institutional ethics committees (Sant’Orsola-Malpighi Hospital, protocol 112/2014/U/Tess and Area Vasta Romagna, protocol 5805/2019) in accordance with the Declaration of Helsinki.

The mutational status of *TP53* was analyzed by SOPHIA Myeloid Solution™ (SOPHiA GENETICS, Switzerland) as previously described [42] or by conventional Sanger sequencing, using the following primers (5′-3′): *TP53* exon 5–6 fw: CACTTGTGCCCTGACTTTCA, rev: TTGCACATCTCATGGGGTTA; *TP53* exon 7–9 fw: CGCACTGGCCTCATCTTGG, rev: TGTCTTTGAGGCATCACTGC. Capillary electrophoresis was performed to analyze *NPM1* mutational status [43] and detect *FLT3*-ITD [44].

Mononuclear cells from bone marrow (BM) or peripheral blood (PB) of 13 newly-diagnosed or relapsed/refractory adult AML patients were collected by density gradient centrifugation using Lymphosep (Biowest, Riverside, MO, USA). Blast percentage was higher than 85%. Cells were cultured in StemSpan™ SFEM-II Medium (STEMCELL Technologies, Vancouver, Canada) containing 2 mM L-Glutamine (GE Healthcare), 20 ng/mL rhIL3, 20 ng/mL FLT3L, 20 ng/mL rhIL-6, 20 ng/mL rhSCF and 20 ng/mL rhG-CSF (PeproTech, London, UK).

### 2.3. Drugs

The MDM2 inhibitor Nutlin-3a (Nut-3a) and the WIP1i GSK2830371 were purchased from Sigma-Aldrich. Compounds were dissolved in DMSO to obtain 10 mM stock solutions and stored at −80 °C (WIP1i) and −20 °C (Nut-3a).

### 2.4. Cell Viability Assay

AML cell lines were seeded in 96-well plates at a concentration of 10,000 cells/well and incubated at 37 °C for 24, 48 and 72 h (h) with increasing drug concentrations: Nut-3a 0.5, 1, 2.5, 5 µM; WIP1i 5, 10, 20 µM (or DMSO, as vehicle). Primary samples were seeded in 6-well plates at concentration of 1 × 10^6^ cells/mL and treated with the highest doses of the two inhibitors (5 and 20 µM for Nut-3a and WIP1i respectively) based on preliminary results from ex vivo experiments (data not shown). Cell line viability was assessed by adding WST-1 reagent (Roche Applied Science, Switzerland) to the culture medium at 1:10 dilution. Cells were incubated at 37 °C and the optical density was measured by Thermo Scientific Multiskan EX microplate ELISA reader at λ450 after 3 h (Thermo Fisher Scientific). Drug effect was expressed as percentage of vehicle-treated cells. IC50 was calculated by GraphPad Prism v6.01 To evaluate synergism, additivity or antagonism of the co-treatment, the combination index (C.I.) was calculated by CompuSyn software (ComboSyn Inc.) [45]. Based on manufacturer’s instructions, we defined: synergism, CI < 1; additivity, CI = 1; antagonism, CI > 1. The viability of 3 AML primary cells upon treatment of Nut-3a and WIP1i was assessed by Trypan Blue staining (Bio-Rad Laboratories, Hercules, CA, USA).

### 2.5. Annexin V-Propidium IODIDE Staining of Apoptotic Cells

AML cell lines and primary cells were treated simultaneously with Nut-3a and WIP1i for 24 or 48 h. Cells were harvested and phosphatidylserine externalization was evaluated using the fluorescein isothiocyanate (FITC) Annexin V Apoptosis Detection Kit (eBioscience™ Thermo Fisher Scientific) according to manufacturer’s instruction. The percentage of apoptotic cells (Annexin V^+^) was determined by flow cytometry (BD Accuri C6 and Facs Canto II Flow Cytometer, BD Biosciences Pharmingen, San Jose, CA USA).

### 2.6. RNA Extraction and Gene Expression Profiling (GEP)

MV-4-11 (*TP53*-wt) and NOMO-1 (*TP53*-mut) cells were treated with Nut-3a (0.5 and 5 µM) and WIP1i (5 and 20 µM), respectively, or the drug combination (or vehicle) for 16 h (h). Cells were harvested and lysed in TRIzol^®^ reagent (Invitrogen, ThermoFisher Scientific). RNA was extracted according to manufacturer’s instructions. Labeled cDNA was prepared and hybridized to the Human Clariom S Assay (ThermoFisher Scientific) following manufacturer’s recommendations. GEP was performed on three independent replicates and analyzed using Transcriptome Analysis Console Software (version 4.0.1, Thermo Fisher Scientific) with signal Space Transformation Robust Multi-Array average (sst-RMA) normalization. Two-fold changes and *p* ≤ 0.05 were selected as thresholds in the supervised data analysis. Gene expression changes induced by the combined treatment were calculated over the vehicle- or the single agent-treated samples. Gene set enrichment analysis (GSEA) was performed by GSEA software (Broad Institute), by comparing single agent exposure or drug combination with vehicle-treated cells. False discovery rate (FDR) ≤ 0.05 was used as cut-off for significance. Gene expression data are available in the Gene Expression Omnibus (GEO) repository, under the accession number GSE156182.

### 2.7. Analysis of Public GEP and RNA-Sequencing Cohorts

GSE6891 [46], GSE13159 [47] and The Cancer Genome Atlas (TCGA) [48] AML data were retrieved from the GEO repository (https://www.ncbi.nlm.nih.gov/gds) and the Genomic Data Commons (GDC) Data Portal (https://gdc.cancer.gov), respectively. Array data were normalized using Transcriptome Analysis Console Software (version 4.0.1) with Robust Multichip Average (RMA) normalization. Read counts from the TCGA dataset were transformed to Counts Per Million (CPM) using calcNormFactors (method = “TMM”) function in edgeR (v3.24.1, R v3.5.1).

### 2.8. Western Blots Analysis

After 16h treatment, 5 × 10^6^ cells from MOLM-13, OCI-AML3, MV-4-11, HEL, KASUMI-1 and NOMO-1 were collected and total protein extracts were prepared in RIPA lysis buffer, containing protease inhibitor cocktail 1X, sodium orthovanadate 1 mM and PMSF 2 mM (Santa Cruz Biotechnology, Dallas, TX, USA). Protein extract concentrations were quantified using the Bicinchoninic Acid (BCA) protein assay kit (Bio-Rad Laboratories). Proteins (30 µg) were loaded on 4–20% Mini-Protean TGX stain-free precast gels (Bio-Rad, Laboratories), blotted on nitrocellulose membranes using a TransBlot Turbo system (Bio-Rad Laboratories) and incubated overnight with primary antibodies, after 1 h blocking in Tris-buffered saline with 0.1% Tween-20 (TBS-T) plus 5% dry milk. The following primary antibodies were used: anti-WIP1 (#SC20712) from Santa Cruz Biotechnology; anti-p53 (PAb 140, NB 200-103) from Novus Biological (Centennial, CO, USA); anti-MDM2 (D1V2Z), anti-p21 WAF1/Cip1 (12D1), all from Cell Signaling (Danvers, MA, USA); anti-β-actin from Sigma-Aldrich. Horseradish peroxidase-conjugated anti-rabbit (NA934) and anti-mouse (NA931) IgG (GE Healthcare) were used as secondary antibodies. β-actin (clone AC-15, Abcam, Cambridge, UK) was used as loading control. The signal was detected using the enhanced chemi-luminescence kit (GE Healthcare) and the ChemiDoc MP system (Bio-Rad Laboratories). Data were analyzed by ImageJ 1.52v software (NIH, Bethesda, MD, USA).

### 2.9. Statistical Analyses

Data are presented as mean ± standard deviation (SD) or median and minimum-to-maximum values for continuous variables, or natural frequencies and percentages for categorical ones.

Normality was assessed by means of the Shapiro-Wilk test. The association between clinical or molecular variables and *PPM1D* expression was performed using Wilcoxon-Mann-Whitney test or the Kruskal Wallis test, as appropriate. When multiple comparisons were performed, p-values adjusted by using the Bonferroni method. One-way analysis of variance (ANOVA) with Dunnett’s post-hoc test wasperformed to compare cell viability, apoptosis and protein expression on multiple groups; Welch *t*-test was used to compare two groups. Statistical analyses were performed using GraphPad 8.0.1 software (GraphPad Inc., San Diego, CA, USA) and R (v3.4.1).

## 3. Results

### 3.1. PPM1D mRNA Levels Differ among Age, Cytogenetic and Mutational Subgroups in AML

To investigate the expression of *PPM1D* across different AML subtypes and its correlation with clinical features, we analyzed 3 independent public transcriptomic cohorts with available clinical (age, disease type) and/or molecular (karyotype, mutations) data. *PPM1D* expression was higher in younger AML patients compared to the elderly (GSE6891, *p* = 0.03; TCGA, *p* = 0.004; Table 1). Moreover, we observed variation in *PPM1D* levels among cytogenetic subgroups (Kruskal Wallis test, TCGA *p* = 0.031; GSE13159, *p* < 0.001, Figure 1A), with higher expression in t(8;21) and inv(16)/t(16;16) cases and lower expression in normal karyotype AML in both the two independent cohorts. In addition, *KMT2A*-rearranged and complex karyotype AML showed high *PPM1D* expression in the TCGA and the GSE13159 datasets, respectively.

We then analyzed *PPM1D* expression according to the mutational status of AML-related genes (Table 1). We did not identify differences according to mutations in *CEBPA*, *DNMT3A*, *KRAS/NRAS*, *IDH1*, *IDH2*, *ASXL1* and *RUNX1*. However, *PPM1D* expression was lower in *FLT3-ITD* AML compared with wt-cases (GSE6891, *p* = 0.017) and in *NPM1*-mut cases compared with wt-ones (GSE6891, *p* = 0.002). Therefore, we classified AML according to the combination of *FLT3*-ITD and *NPM1* mutations and evaluated *PPM1D* expression across the subgroups (Kruskal Wallis test, GSE6891 *p* = 0.009). We observed that *NPM1*-mut AML displayed lower *PPM1D* expression, irrespective of *FLT3* mutational status (Figure 1B). AML cases carrying *TP53* mutations had lower *PPM1D* levels (Kruskal Wallis test, *p* = 0.007) in the TCGA dataset (*TP53* mutational data were not available from the other cohorts).

We then analyzed WIP1 protein expression in a panel of *TP53*-wt (MV-4-11, OCI-AML3 and MOLM-13) and *TP53*-mut cells (NOMO-1, KASUMI-1 and HEL). In line with data from public cohorts, we observed high WIP1 levels in one of the *KMT2A*-rearranged models (MOLM-13) and in the t(8;21) KASUMI-1 cell line, intermediate levels in *NPM1*-mut OCI-AML3 cells and very low expression in the other *TP53*-mut models (NOMO-1 and HEL, Appendix A).

Overall, these data indicate that *PPM1D* expression is heterogeneous in AML and it varies according to cytogenetic and molecular status.

### 3.2. Combined Inhibition of Nut-3a and WIP1i Synergistically Reduces AML Cells Viability

To investigate whether p53 activation via simultaneous inhibition of WIP1 and MDM2 may be a valuable therapeutic strategy in AML, we performed in vitro preclinical assays. Single agent Nut-3a did not reduce cell viability in the *TP53*-mut cells, while showing a time and dosage-dependent effect in all the *TP53*-wt cells (Figure 2A), with IC50 values below 1 µM at 72 h (Figure 2B). We observed a cell viability reduction at 72 h of 97.4% and 99.3% (MV-4-11, *p* < 0.01), of 88.9% and 99.4% (MOLM-13, *p* < 0.001), of 23% and 40.3% (OCI-AML3, *p* < 0.01) at 2.5 and 5 µM of Nut-3a drug concentrations, respectively. OCI-AML3, along with MV-4-11, showed a better response to single agent WIP1i, with a significant decrease of cell viability at 72 h: 52.2% and 57.6% (OCI-AML3, *p* < 0.01); 37.9% and 78.2% (MV-4-11, *p* < 0.001) at the highest doses (10 and 20 µM, respectively, Figure 2C). WIP1i as single agent did not significantly affected MOLM-13, HEL, KASUMI-1 and NOMO-1 cell viability (Figure 2B,C).

We then tested the efficacy of the drug combination by incubating AML cell lines with Nut-3a and WIP1i for 24, 48 and 72 h. The combined treatment reduced the viability of *TP53*-wt cells (Figure 2D), while sparing the *TP53*-mut ones that remained insensitive (Figure 2E). The combination index analyses showed a synergic (or additive, according to dosages) effect of Nut-3a and WIP1i combination in MOLM-13, MV-4-11 and OCI-AML3, especially using low Nut-3a doses (Appendix A).

### 3.3. WIP1i Sensitizes TP53-wt AML Cells to Nut-3a-Induced Apoptosis

To further investigate the mechanism of action of the drug combination, induction of apoptosis was evaluated. Based on the combination index analysis, cell lines were treated with different concentrations of Nut-3a and WIP1i (2.5 and 20 µM for OCI-AML3; 0.5 and 5 µM for MV-4-11; 0.5 and 10 µM for MOLM-13; 5 and 20 µM for HEL, KASUMI-1 and NOMO-1, respectively) for 24 and 48 h. We detected a significant increase in the percentage of apoptotic cells in the *TP53*-wt MOLM-13 and MV-4-11 cell lines, when simultaneously treated with the two drugs, compared with vehicle or single agent exposure at 48 h (Annexin-V^+^ MOLM-13 cells: drug combination, 26 ± 1%, Nut-3a, 12 ± 5.3%, WIP1i, 8.7 ± 2%, DMSO, 7.1 ± 2.5%; Annexin-V^+^ MV-4-11 cells: drug combination, 41.8 ± 4.7% Nut-3a, 17.4 ± 1.9%, WIP1i, 12.3 ± 1.7%, DMSO 6.2 ± 1.5%; Figure 3A). OCI-AML3 showed an enhanced apoptotic response to the combined treatment when compared with WIP1i alone or vehicle and a trend towards increased apoptosis compared with Nut-3a as single agent (Figure 3A). In line with the cell viability results, apoptosis of NOMO-1, HEL and KASUMI-1 cells were barely affected by the combined treatment at 48 h (18.9 ± 11.7%, 11.3 ± 3.8% and 17.1 ± 2.6% of Annexin-V^+^ cells upon drug combination vs. Nut-3a: 13.2 ± 3.8%, 9.9 ± 0.9% and 10.9 ± 2.2%; WIP1i: 9.6 ± 4.2%, 6.6 ± 4.1% and 11.8 ± 1.1%; vehicle: 9.3 ± 4.4%, 6 ± 3.2% and 8.9 ± 0.1% for NOMO-1 and HEL, respectively, Figure 3B).

These results were validated in primary *TP53*-wt AML cells (Appendix A) in ex vivo assays. After 48 h, the combination of WIP1i and Nut-3a induced a significant decrease of cell viability compared to single agents or vehicle treatment (Figure 3C). This was accompanied by a progressive increase of apoptotic cells in the combined treatment at 48 h (45.2 ± 14.6%, compared with 35 ± 18.4% of Nut-3a, 27 ± 11.5% of WIP1i-treated samples and 21.8 ± 8.2% of control cells, Figure 3D). Of note, *NPM1*-mut AML showed a higher ex vivo sensitivity to Nut-3 and to the combined treatment compared with *NPM1*-wt cells at 48 h (Nut-3a: *p* = 0.020; drug combination: *p* = 0.040, Figure 3E). Conversely, primary *TP53*-mut leukemic cells neither responded to single agent nor to the combined treatment (Figure 3F).

### 3.4. The Inhibition of WIP1 and MDM2 Altered the Expression of p53 Pathway-Related Genes in TP53-wt Cells

To elucidate changes in the transcriptional program of AML cells induced by Nut-3a and WIP1i combination, we performed gene expression analysis on representative *TP53*-wt and *TP53*-mut cell lines (MV4-11 and NOMO1) after 16h of treatment. This time point allowed the evaluation of transcriptional changes (Appendix A), avoiding excessive cell death. The analyses of differentially expressed genes showed enrichment of a p53 signature in MV-4-11 cells treated with the drug combination (vs. control, Normalized Enrichment Score (NES) = 2.32, FDR ≤ 0.001 Figure 4A) and in those treated with either Nut-3a or WIP1i as single agents (vs. vehicle (Nut-3a vs. DMSO: NES = 2.13, FDR ≤ 0.001; WIP1i vs. DMSO: NES = 1.49, FDR = 0.02, Appendix A). Significantly upregulated genes belonging to this signature or annotated as bona fide p53 targets included *MDM2*, *CDKN1A*, *PLK2*, *GADD45A*, *IER5* (Figure 4B). Overall, 33 genes from the signature or annotated as *bona fide* p53 targets [49] were upregulated in MV-4-11 cells, while only 3 genes showed increased expression in NOMO-1 cells treated with the drug combination (vs. control, Appendix A).

The protein level p53 and of key signature genes (*MDM2* and *CDKN1A*) was evaluated in the whole panel of cell lines, along with WIP1. The overexpression of MDM2 and p21 was confirmed by immunoblot analysis in MOLM-13, OCI-AML3 and MV-4-11 cells (single agents and/or combination, Figure 4C and Appendix A). Moreover, p53 protein levels were increased by all drug treatments in the *TP53*-wt cells. Conversely, in the *TP53*-mut models, p53 protein was barely detectable in NOMO-1 cells and was not significantly altered by the treatments also in HEL and KASUMI-1 (Figure 4D and Appendix A). Moreover, the p53 pathway was not affected in the *TP53*-mut cell lines, except for MDM2 downregulation in KASUMI-1 cells under WIP1i pressure (and a trend in NOMO1 cells when exposed to the drug combination) and WIP1 reduction in NOMO-1 treated cells. Finally, WIP1 protein showed a trend towards increased expression after Nut-3a treatment in the *TP53*-wt cells, (Figure 4C and Appendix A), while being downmodulated by the drugs and their combination in NOMO1 cells (Figure 4D and Appendix A).

### 3.5. GSEA Analysis Showed the Enrichment of Single Agent- and Drug Combination-Specific Genes and Pathways in MV-4-11 and NOMO-1 Cells

Additional treatment-specific pathways were enriched upon drug exposure in the two cell lines selected for GEP analyses (Table 2), suggesting other mechanistic information, with differences between *TP53*-wt and *TP53*-mut cells. Single agent treatment with WIP1i in MV-4-11 showed enrichment of MYC targets (e.g., upregulation of *SRSF1, RUVBL2, NIP7, PHB* and downregulation of *PSMC4*) and unfolded protein response (e.g., upregulation of *CXXC1* and *CALR*, Figure 5A). Gene expression changes induced by Nut-3a treatment led to enrichment of the following drug-specific signatures in MV-4-11: PI3K-mTOR signaling (e.g., upregulated *MAP3K7*, *RAC1*, *MAPKAP1*, *PPP1CA*), which can be involved in cell cycle and metabolism regulation, as oligosaccharide-lipid intermediate biosynthetic process (upregulated *ALG3*, *MPDU1*); positive regulation of cysteine-type endopeptidase activity involved in apoptotic signaling (upregulated *TNFRSF10B*, *BAX*, Figure 5B). In MV-4-11 cells, combined Nut-3a and WIP1i treatment, but not single agent exposure, led to enrichment of signatures of TNFA signaling via NFKB (e.g., *TNFSF9, GEM* upregulated); interferon-γ response (e.g., *PNPT1*, *BANK1* upregulated) and interferon-α response (*PNPT1* and *RNF31* upregulated, Figure 5C), suggesting the induction of an inflammatory status.

In NOMO-1 cells, WIP1 inhibition had mild effects, with no enriched gene sets compared with control cells. Conversely, MDM2 inhibition and the drug combination led to upregulation of inflammatory signatures response (e.g., *IL7R* and *TNFRSF9* upregulated) and TNFA signaling via NFKB (e.g., *IER3*, *PHLDA1* upregulated, Appendix A and Table 3).

## 4. Discussion

*PPM1D* is an emerging oncogene strictly related to the p53 pathway and involved in clonal hematopoiesis and AML pathogenesis [25,28,29,50]. WIP1, the protein encoded by *PPM1D*, is overexpressed in many solid tumors, conferring a poor prognosis [16,51,52], while its deficiency causes defects in hematopoietic differentiation, immune system and inflammation [53]. To better understand the functional role of WIP1 in AML and explore its therapeutic potential in drug combinations, we combined the analysis of gene expression datasets and in vitro/ex vivo preclinical data.

Although a recent study suggested that *PPM1D* may be highly expressed in poor risk AML cases [54], our results from at least 2 independent cohorts, showed high *PPM1D* levels in core binding factor AML and heterogeneous levels in cases with high risk and complex karyotype. Moreover, patients carrying *TP53* mutations were characterized by lower *PPM1D* expression compared with wt-ones. Since a feedback-regulatory loop regulates *PPM1D* and p53, an impaired p53 activity, which is a feature of complex karyotype AML [35,36], may result in *PPM1D* expression changes.

*PPM1D* expression was also lower in *NPM1*-mut patients compared with *NPM1*-wt cases. *PPM1D* is one of the main actors in nucleolar formation through sequential phosphorylation of *NPM1* [55]. Phosphorylation of cytoplasmic *NPM1* at the Threonine 199 residue is important during mitotic progression, by preventing centrosome reduplication, thus reducing genotoxic stress conditions that may activate *PPM1D* expression [55,56]. In our ex vivo tests, *NPM1*-mutated AML cells showed an increased sensitivity to Nut-3a and the drug combination compared with *NPM1*-wt cases (excluding *TP53*-mut patients), suggesting a potential vulnerability.

Several studies have proven the efficacy of the MDM2 inhibitor Nut-3a in combination with chemotherapeutic and targeted agents [38,39,40]. Here, we demonstrated that the pharmacological inhibition of WIP1 synergizes with Nut-3a in *TP53*-wt AML cells promoting a significant induction of apoptosis (Figure 6A,B). The comparison between combined and single agent treatments highlighted a significant decrease of cell viability in MOLM-13, MV-4-11 and OCI-AML3 models and a strong induction of apoptosis. The *NPM1*-mut OCI-AML3 cell line showed a good response to WIP1i alone and a high percentage of apoptosis when treated with the drug combination. On the contrary, *TP53*-mut HEL, KASUMI-1 and NOMO-1 cells were insensitive to the tested treatments, as confirmed in primary samples. Gene and protein expression changes induced by drug treatment of the *TP53*-wt cell lines revealed a cooperation between Nut-3a and WIP1i in the stabilization of p53 protein level and activity, with activation of a p53 signature and upregulation of its downstream partners, including p21 and MDM2. In the resistant cells p53 and p21 proteins were not affected by the treatments, while WIP1 and MDM2 showed cell line-dependent changes, the latter being in contrast with the results obtained in the sensitive models. Gene expression signatures analysis highlighted additional pathway alterations induced by Nut-3a or WIP1i single agent and combined treatments. In particular, MDM2 inhibition led to enrichment of inflammatory signatures response- and NFKB-related genes in *TP53*-mut cells that was observed upon combined treatment in the *TP53*-wt cells [57]. These changes may alter the crosstalk between leukemic cells and the microenvironment and provide the rationale for novel drug combinations acting on the malignant cells and the immune response.

Overall, based on our results we hypothesize that the upregulation of MDM2 and the induction of MYC-related signatures mediated by WIP1 inhibition in *TP53*-wt cells may enhance cell sensitivity to Nut-3a [58,59]. MDM2 is also a potential marker of response to the drug combination, as previously observed for Nut-3a treatment [41]. Furthermore, our data suggest that, at least in some models, the combined treatment can stabilize the p53-phosphorylated form, which in turn is able, in cells expressing wildtype p53, to propagate the stress-mediated response by activating downstream targets and triggering apoptosis (Figure 6A,B) [60].

In conclusion, here we proposed a novel therapeutic strategy for *TP53*-wt AML based on the synergistic combination of Nut-3a and WIP1i and unraveled the transcriptomic changes induced by WIP1i alone or in combination with Nut-3a in AML cells. Future in vivo studies are needed to confirm these preclinical data and test the toxicity of the combined WIP1 and MDM2 inhibition. A large fraction of AML cases, in particular aneuploid and complex karyotype, has structurally intact but dysfunctional p53. In these patients, WIP1 inhibition may potentially improve the efficacy of novel experimental agents.

## Figures and Tables

**Figure 1 biomedicines-09-00388-f001:**
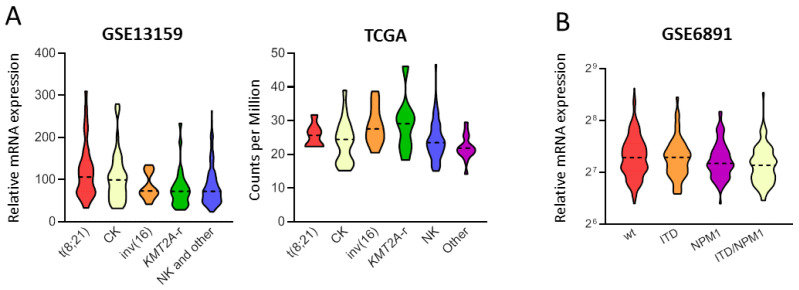
*PPM1D* expression in AML according to cytogenetic classification and *FLT3*-ITD/*NPM1* status. (**A**) Violin plot of *PPM1D* transcript level in the GSE13159 and TCGA cohorts showing a significant different distribution among cytogenetic subgroups (t(8;21); CK: complex karyotype; inv(16): inv(16)/t(16;16); *KMT2A*-r: *KMT2A*-rearranged; NK: normal karyotype; AML carrying NK and other cytogenetics abnormalities could not be distinguished in GSE13159 due to data unavailability, thus named as “NK and other”). (**B**) Violin plot of *PPM1D* transcript level in the GSE6891 cohort showing a significant different distribution among molecular subgroups defined by *FLT3*-ITD and *NPM1* mutations (wt: *FLT3*-ITD−/*NPM1*-wt; ITD: *FLT3*-ITD+/*NPM1*-wt; NPM1: *FLT3*-ITD−/*NPM1*-mut; ITD/*NPM1*: *FLT3*-ITD+/*NPM1*-mut). The plots represent the frequency distribution of *PPM1D* levels (from minimum to maximum) and the dotted line indicates the median value (only cohorts showing statistically significant results are reported in the figure).

**Figure 2 biomedicines-09-00388-f002:**
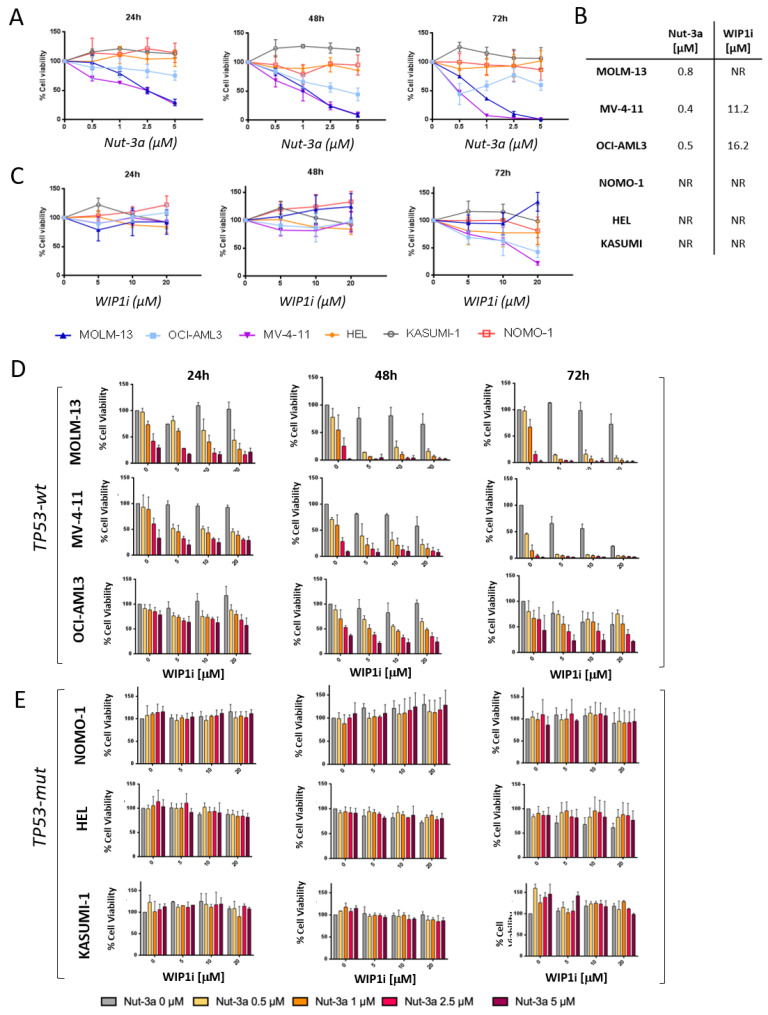
Viability of AML cell lines treated with Nut-3a and/or WIP1i. (**A**) Percentage of viable MOLM-13, MV-4-11, OCI-AML3, NOMO-1, HEL and KASUMI-1 AML cells treated with increasing concentrations of single agent Nut-3a (from 0.5 to 5 μM) for 24, 48 and 72 h. (**B**) IC50 values of AML cell lines at 72 h of treatment with Nut-3a or WIP1i (NR = not reached). (**C**) Percentage of viable cells treated with increasing concentrations of single agent WIP1i (from 5 to 20 μM) for 24, 48 and 72 h. Inhibition of cell viability induced in *TP53*-wt (**D**) and *TP53*-mut (**E**) AML cell lines by the combination of increasing concentrations of Nut-3a (from 0.5 to 5 μM) and WIP1i (from 5 to 20 µM) at 24, 48 and 72 h. Average value and standard deviation of 3 independent experiments are shown.

**Figure 3 biomedicines-09-00388-f003:**
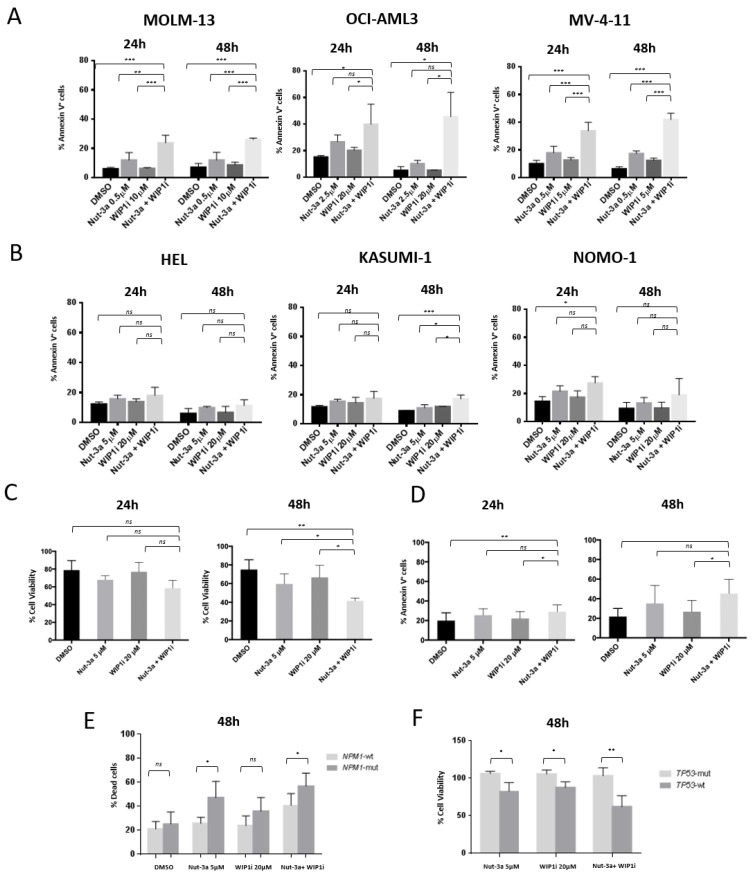
Apoptotic response of AML cell lines and primary cells to combined Nut-3a and WIP1i treatment. (**A**) Histograms showing the percentage of apoptotic (AnnexinV^+^ cells) cells in *TP53*-wt (**A**) and *TP53*-mut cell lines (**B**) after 24 and 48 h treatment with single and combined Nut-3a and WIP1i. Average value and standard deviation of 3 independent experiments are shown. (**C**) Cell viability and (**D**) apoptotic response of *TP53*-wt AML primary cells (*n* = 3 and *n* = 6, respectively) after 24 h and 48 h treatment with single and combined Nut-3a and WIP1i treatments. (**E**) Histograms showing the percentage of dead cells in *NPM1*-mut and *NPM1*-wt primary AML cells (*n* = 5 each) after 48 h treatment with single and combined Nut-3a and WIP1i. (**F**) Histograms showing the percentage of viable cells (normalized on vehicle-treated cells) in *TP53*-mut (*n* = 3) and *TP53*-wt (*n* = 4) primary AML cells after 48 h treatment with single and combined Nut-3a and WIP1i (* *p* < 0.05, ** *p* < 0.01, *** *p* < 0.001, ns: not significant).

**Figure 4 biomedicines-09-00388-f004:**
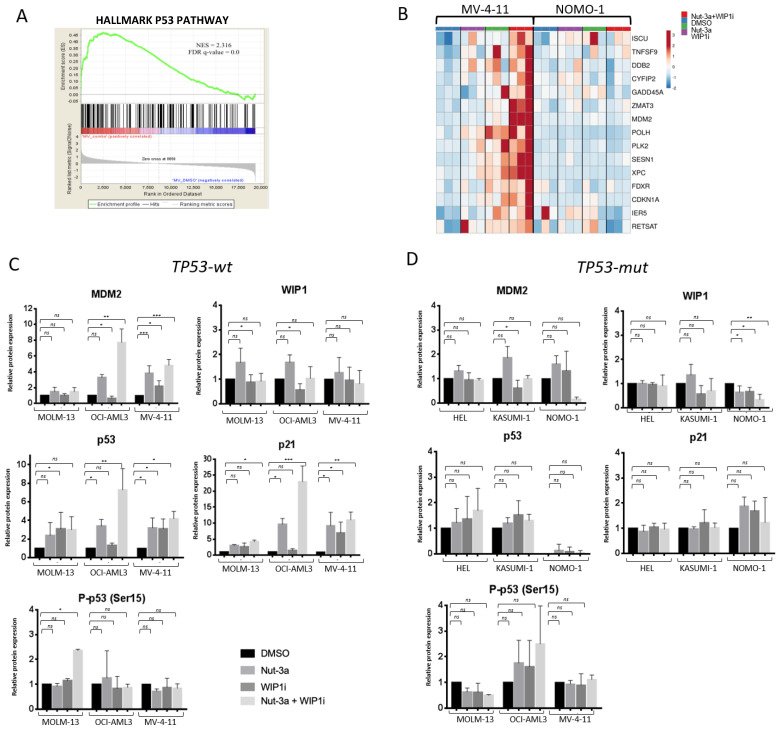
Changes in the expression of p53-related genes induced by the treatment in *TP53*-wt and *TP53*-mut cells. Cells were harvested after 16 h of treatment (Nut-3a 0.5 and 5 µM; WIP1i 5 and 20 µM, for *TP53*-wt and *TP53*-mut cells, respectively) both for gene expression microarray and protein analyses. (**A**) Enrichment of p53 signature in MV-4-11 cells treated with the drug combination vs. vehicle (gene expression microarray). (**B**) Heatmap of MV-4-11 and NOMO-1 cells showing the significantly deregulated genes (differential expression analysis between Nut-3a+WIP1i-treated and vehicle-treated MV-4-11 cells, fold change ≥ 2, *p* < 0.05) belonging to the p53 signature in the analyzed models. (**C**) Protein quantification of p53-related genes in treated *TP53*-wt and (**D**) *TP53*-mut cells. Histograms show the average value of 3 independent experiments ± SD (* *p* < 0.05, ** *p* < 0.01, *** *p* < 0.001, ns: not significant).

**Figure 5 biomedicines-09-00388-f005:**
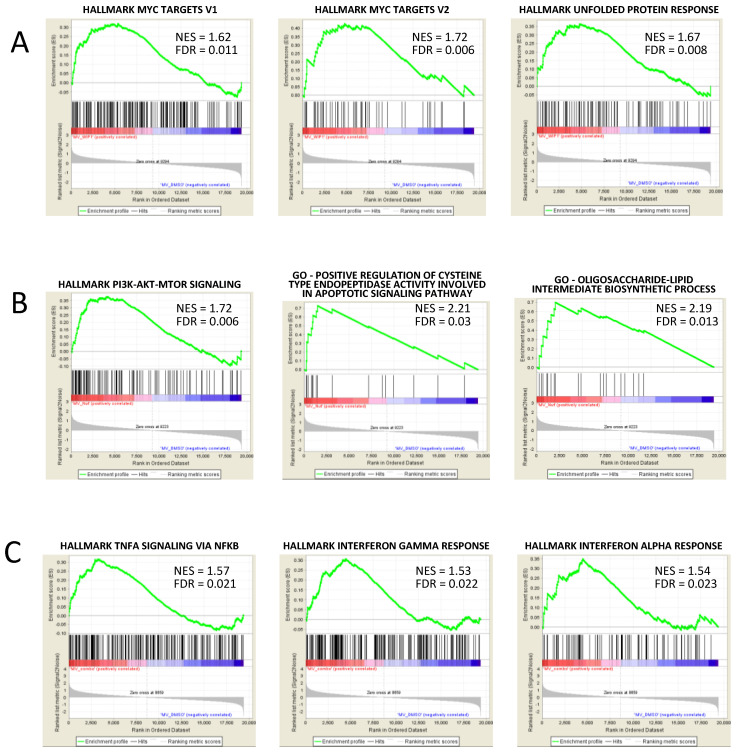
Treatment-specific transcriptional signatures enriched in MV-4-11 and NOMO-1 cells. GSEA plots of significantly enriched signatures in MV-4-11 cells treated with (**A**) WIP1i, (**B**) Nut-3a, or (**C**) their combination vs. control cells (NES: normalized enrichment score, FDR: false discovery rate).

**Figure 6 biomedicines-09-00388-f006:**
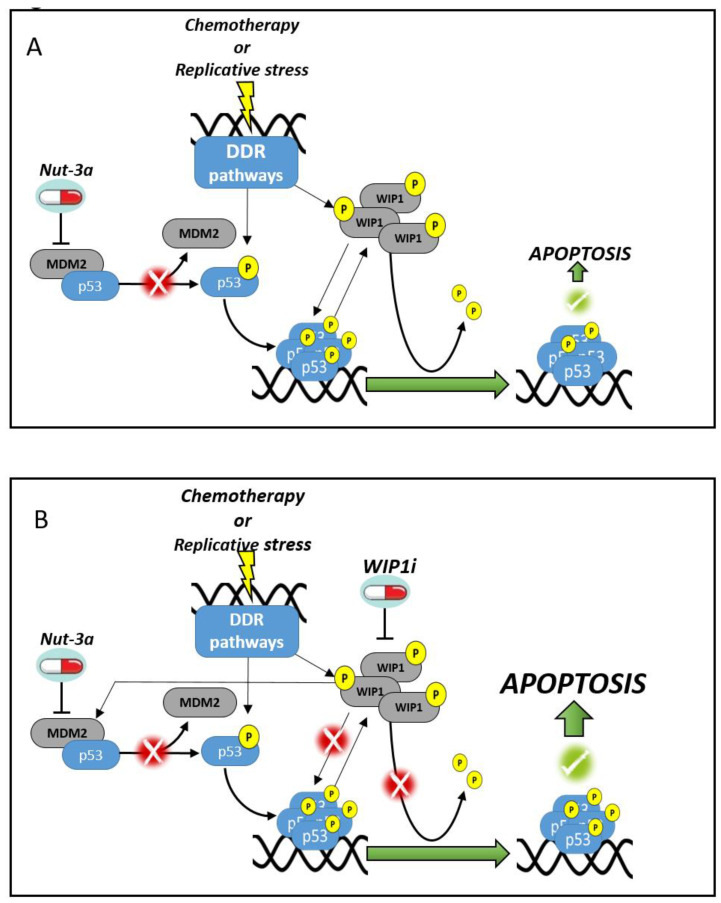
Proposed mechanism of action of Nut-3a and WIP1i combined treatment in AML cells. MDM2 inhibition enhances p53-dependent response to DNA damages induced by chemotherapy agents or replicative stress. Once Nut-3a binds to MDM2, p53 is released and activated through phosphorylation. Active p53 promotes the induction of apoptosis. (**A**) WIP1 is involved in the regulation of response to Nut-3a and dephosphorylates p53. WIP1 and p53 are co-regulated by a feedback-loop. (**B**) When WIP1i is simultaneously added to Nut-3a, p53 activation is enforced, resulting in enhanced apoptosis of AML cells. The arrows represent a stimulatory signal, truncated arrows represent a inhibition signal.

**Table 1 biomedicines-09-00388-t001:** Association between PPM1D expression levels and clinical/molecular data across public datasets.

Variable	GSE6891 [48] (*n* = 499)			GSE13159 [49] (*n* = 458)			TCGA [50] (*n* = 178)		
	*PPM1D* Median	*n* (%)	*p*-Value	*PPM1D* Median	*n* (%)	*p*-Value	*PPM1D* Median (Min–Max)	*n* (%)	*p*-Value
	(Min–Max)			(Min–Max)					
Age ^‡^									
<60-years	151.6 (83.9–393.3)	417 (84.8)	0.03	*NA*	*NA*	*NA*	25.6 (15.1–46.7)	92 (51.7)	0.004
≥60-years	146.7 (94.5–234.4)	75 (15.2)					22.6 (14.2–39.4)	86 (48.3)	
		*NA* = 7							
Cytogenetic group ^‡^									
t(8;21)	169.8 (89.9–318.9)	38 (9.0)	0.185	106.2 (32.6–309.9)	35 (7.6)	<0.001	25.7 (22.3–31.7)	7 (4.2)	0.031
inv(16)/t(16;16)	156.3 (95.8–319.7)	42 (10.0)		73.2 (41.4–134.6)	27 (5.9)		27.6 (20.5–38.7)	11 (6.6)	
NK	147.5 (83.9–371.3)	171 (40.5)		*NA*	0		23.5 (15.1–46.7)	96 (57.8)	
CK	146.4 (97.9–259.7)	34 (8.1)		99.3 (31.5–279.4)	45 (9.8)		24.4 (15.2–39.0)	21 (12.7)	
*KMT2A*-r	139.6 (94.3–256.3)	17 (4.0)		72.2 (28.2–233.4)	29 (6.3)		29.1 (18.3–46.1)	10 (6.0)	
Other	143.1 (84.1–393.3)	120 (28.4)		*NA*	0		21.9 (14.2–29.5)	21 (12.7)	
Normal/Other *				72.2 (23.4–263.2)	322 (21.8)				
		*NA* = 77						*NA* = 12	
*FLT3*-ITD ^‡^									
*FLT3*-ITD^+^	146.1 (87.6–371.3)	134 (26.9)	0.017	*NA*	*NA*	*NA*	24.2 (14.2–46.7)	32 (18.4)	0.87
*FLT3*-ITD^−^	152.2 (83.9–393.3)	365 (73.1)					23.2 (15.2–39.0)	143 (81.7)	
								*NA* = 3	
*NPM1* status ^‡^									
*NPM1*-mut	143.1 (83.9–371.3)	159 (39.9)	0.002	*NA*	*NA*	*NA*	23.7 (16.1–46.7)	53 (30.3)	0.69
*NPM1*-wt	155.8 (84.1–393.3)	340 (68.1)					24.0 (14.2–46.1)	122 (69.7)	
								*NA* = 3	

^‡^ The sum does not add up to the total due to missing values. * “NK” and “other” cytogenetic subgroups were not distinguished in GSE13159. CK: complex karyotype; ITD: internal tandem duplication; *KMT2A*-r: *KMT2A*-rearranged; min-max: minimum-to-maximum value; mut: mutated; NA: not available; NK: normal karyotype; wt: wildtype.

**Table 2 biomedicines-09-00388-t002:** Most significant enriched pathways in MV-4-11 cells upon drug treatment.

Pathway Name	NES	FDR	Reatment Comparison
Hallmark of p53 Pathway	2.31	≤0.001	Nut3a+WIP1i vs. DMSO
	2.13	≤0.001	Nut3a vs. DMSO
	1.49	0.029	WIP1i vs. DMSO
Hallmark of Protein Secretion	1.98	≤0.001	Nut3a+WIP1i vs. DMSO
	2.24	≤0.001	Nut3a vs. DMSO
	1.73	0.005	WIP1i vs. DMSO
Hallmark of Oxidative Phosphorylation	1.87	0.001	Nut3a+WIP1i vs. DMSO
	1.94	≤0.001	Nut3a vs. DMSO
	2.31	0.005	WIP1i vs. DMSO
Hallmark of Apoptosis	1.70	0.009	Nut3a+WIP1i vs. DMSO
	1.70	0.003	Nut3a vs. DMSO
	1.52	0.024	WIP1i vs. DMSO
Hallmark of mTORC1 Signaling	1.75	0.002	Nut3a vs. DMSO
	1.88	0.003	WIP1i vs. DMSO
Hallmark of Fatty Acid Metabolism	1.75	0.002	Nut3a vs. DMSO
	1.62	0.010	WIP1i vs. DMSO
Hallmark of DNA Repair	1.58	0.014	Nut3a vs. DMSO
	1.63	0.011	WIP1i vs. DMSO
Hallmark of Peroxisome	1.52	0.024	Nut3a vs. DMSO
	1.80	0.005	WIP1i vs. DMSO
Hallmark of TNFA Signaling Via NFKB	1.57	0.021	Nut3a+WIP1i vs. DMSO
Hallmark of Interferon Gamma Response	1.53	0.022	Nut3a+WIP1i vs. DMSO
Hallmark of Interferon Alpha Response	1.54	0.023	Nut3a+WIP1i vs. DMSO
Hallmark of PI3K AKT mTOR Signaling	1.73	0.002	Nut3a vs. DMSO
Positive Regulation Of Cysteine Type Endopeptidase Activity Involved In Apoptotic Signaling Pathway	2.21	0.030	Nut3a vs. DMSO
Oligosaccharide-Lipid Intermediate Biosynthetic Process	2.19	0.013	Nut3a vs. DMSO
Hallmark of MYC Targets_V2	1.72	0.006	WIP1i vs. DMSO
Hallmark of Unfolded Protein Response	1.67	0.008	WIP1i vs. DMSO
Hallmark of MYC Targets_V1	1.62	0.011	WIP1i vs. DMSO

FDR: false discovery rate; NES: normalized enrichment score.

**Table 3 biomedicines-09-00388-t003:** Most significant enriched pathways in NOMO-1 cells upon drug treatment.

Pathway Name	NES	FDR	Treatment Comparison
Hallmark of Inflammatory Response	1.99	0.001	Nut3a+WIP1i vs. DMSO
	1.53	0.043	Nut3a vs. DMSO
Hallmark of TNFA Signaling Via NFKB	2.02	0.002	Nut3a+WIP1i vs. DMSO
	1.52	0.022	Nut3a vs. DMSO

FDR: false discovery rate; NES: normalized enrichment score.

## Data Availability

Gene expression data are available in the Gene Expression Omnibus (GEO) repository, under the accession number GSE156182.

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
