# Peer review of "Pharmacological Inhibition of WIP1 Sensitizes Acute Myeloid Leukemia Cells to the MDM2 Inhibitor Nutlin-3a"

_biomedicines, 2021, doi:10.3390/biomedicines9040388_

Round 1

Reviewer 1 Report

The authors addressed the requested changes.

Author Response

We thank the reviewer.

Reviewer 2 Report

In the paper Pharmacological inhibition of WIP1 sensitizes acute myeloid leukemia cells to the MDM2 inhibitor Nutlin-3a, Fontana et al. studied the in vitro effects of the combinations of WIP1 inhibitor and MDM2 inhibitor in TP53-wt and TP53-mut AML cell lines. The authors evaluated cells viability and apoptosis in 6 AML cell lines and in 9 primary AML samples after this pharmacological treatment. Moreover, they performed gene expression analysis and immunoblot in order to identify pathways mediated by these drugs.

The paper is well written. Nutlin-3a treatment is well known in AML, while the combination of MDM2 and WIP1 inhibitors has been studied in other solid tumors cells. On the whole, the results are interesting but the mechanistic model proposed is not well supported and argued by in silico results. They demonstrated that p53 pathway is increased after treatments in TP53-wt cells, but other gene expression results are a little bit speculative and not well supported by other experiments (e.g. western blot) in immortalized or primary cells.  The experimental setting is based mainly on in vitro experiments. Moreover, it is not clear which patients could benefit of this pharmacological approach in a hypothetic clinical trial because PPM1D expression investigation showed heterogeneous results.

These are my comments:

Major comments

  • In my opinion the discussion part about PPM1D expression investigation across AML subtypes in 3 different cohorts is not very informative. This analysis did not identify any particular AML subtype that could take advantage of WIP1 inhibitor due to its high PPM1D expression. The results are heterogeneous and it seems that there is no correlation between PPM1D expression and prognosis.

Moreover also in the cell lines you did not show the basal expression of PPM1D to identify cells that could respond better to WIP1 inhibitor. Please add to the results and, if possible, reduce this discussion part.

  • In the text you affirm that AML cases carrying TP53 mutations had lower PPM1D levels but in table 1 the median is higher in TP53mut vs TP53-wt patients. Please put the correct information.
  • I appreciated that you performed experiments in 6 cell lines in order to minimize heterogeneity but HEL and Kasumi-1 cell lines show homozygosity for their TP53 mutations. In this situation, drugs stabilize only mutated p53 proteins and maybe it could be the reason for TP53 mutated cells poor response. Moreover, primary cells were isolated from patients with no TP53 mutations, in this context it is difficult to affirm that patients in vivo couldn’t benefit of this combinatorial approach. If it possible to obtain additional TP53-mut primary samples it could be interesting study their behavior after treatment.
  • In the discussion you affirm that primary AML cells NPM1 mutated respond to treatment. But, from the results it seems that all ex-vivo cells respond to treatment independently of NPM1 mutation. Did you compare cell viability between NPM1-mut vs NPM1-wt patients? And for other mutations?
  • I did not understand the utility of PPM1D co-expression analysis in section 3.1. It is only descriptive and it is useless to the following analyses and discussions. Please motivate this section or remove it.
  • It is not clear which genes were used to make heatmap in fig4. These genes are differentially expressed between Nut-3a+WIP1i-treated and DMSO treated TP53-wt cell and they are also involved in p53 pathway? Is it correct? Which is the message of this result? Are there other p53 targets that are expressed in TP53-mut but not in TP53-wt?
  • In the discussion you proposed that “the combined treatment may stabilize the p53-phosphorylated form, which in turn is able to propagate the stress mediated response by activating downstream targets and triggering apoptosis”. Did you check phospho-p53 in western blot after treatment?

Minor Comments

  • In section 2.4 you affirm “the viability of 3 AML primary cells”, but in supplementary are 9 samples.
  • Please set a correct Table 1 formatting. Variables and data sometimes are not in the same rows.
  • In the barplot in fig 3 and 4, sometimes DMSO bar is black and sometimes is grey. Please uniform graphs.

Author Response

Major comments

  • In my opinion the discussion part about PPM1D expression investigation across AML subtypes in 3 different cohorts is not very informative. This analysis did not identify any particular AML subtype that could take advantage of WIP1 inhibitor due to its high PPM1D expression. The results are heterogeneous and it seems that there is no correlation between PPM1D expression and prognosis. Moreover also in the cell lines you did not show the basal expression of PPM1D to identify cells that could respond better to WIP1 inhibitor. Please add to the results and, if possible, reduce this discussion part.

Reply. We agree with the reviewer on the fact that PPM1D expression does not correlate with prognosis in our analysis. This may be due to the fact that PPM1D is highly expressed in good prognosis cytogenetic subgroups (as core binding factor AML) but is also expressed at intermediate level by complex karyotype AML. We analyzed basal WIP1 expression in AML cell lines (Figure S1) and we show that KASUMI-1 cells (t(8;21)) express high WIP1 levels, along with one of the KMT2A-rearranged model (MOLM-13), in line with data obtained in primary AML samples. Moreover, the NPM1-mut OCI-AML3 cells showed intermediate levels and the other TP53-mut cell lines (HEL and NOMO-1) express very low WIP1 protein. The response to WIP1 inhibition is likely affected by multiple factors and their molecular background is a relevant one. In the revised manuscript we show that TP53-mut AML respond neither to single agent treatments, nor the drug combination (Fig. 3F). Conversely, among TP53-wt cases, NPM1-mut ones showed a better ex vivo response to Nut-3a and to the drug combination compared with NPM1-wt cases (Fig. 3E). This new data provide the connection between PPM1D expression investigation across AML subtypes and in vitro/ex vivo tests.

  • In the text you affirm that AML cases carrying TP53 mutations had lower PPM1D levels but in table 1 the median is higher in TP53mut vs TP53-wt patients. Please put the correct information.

Reply. We apologize with the reviewer. Primary AML cases carrying TP53 mutations have lower PPM1D levels. We have corrected the information in Table 1.

  • I appreciated that you performed experiments in 6 cell lines in order to minimize heterogeneity but HEL and Kasumi-1 cell lines show homozygosity for their TP53 mutations. In this situation, drugs stabilize only mutated p53 proteins and maybe it could be the reason for TP53 mutated cells poor response. Moreover, primary cells were isolated from patients with no TP53 mutations, in this context it is difficult to affirm that patients in vivo couldn’t benefit of this combinatorial approach. If it possible to obtain additional TP53-mut primary samples it could be interesting study their behavior after treatment.

Reply. We agree with the reviewer on the fact that in the HEL and KASUMI-1 cell lines, the treatment can stabilize the mutated protein that is not able to activate the downstream targets, due to homozygous mutations. We have tested the ex vivo efficacy of the drug combination on TP53-mutated primary samples (n=3). The data show that TP53-mut cases are neither sensitive to the single agent treatments nor to the drug combination (Fig. 3F).

  • In the discussion you affirm that primary AML cells NPM1 mutated respond to treatment. But, from the results it seems that all ex-vivo cells respond to treatment independently of NPM1 mutation. Did you compare cell viability between NPM1-mut vs NPM1-wt patients? And for other mutations?

Since our data on pubic cohorts showed differential expression of PPM1D according to NPM1 and TP53 mutational status, in the revised manuscript we compared the ex vivo sensitivity of TP53-mut versus TP53-wt AML and of NPM1-mut versus NPM1-wt AML. We show that TP53-mut cases do not respond to treatment (Fig. 3F). Conversely, TP53-wt cases are sensitive to the treatment and among them, NPM1-mut AML showed a better ex vivo response to Nut-3a and to the drug combination compared with NPM1-wt cases (Fig. 3E).

  • I did not understand the utility of PPM1D co-expression analysis in section 3.1. It is only descriptive and it is useless to the following analyses and discussions. Please motivate this section or remove it.

Reply. According to the reviewer’s suggestion, we have removed this paragraph from the manuscript.

  • It is not clear which genes were used to make heatmap in fig4. These genes are differentially expressed between Nut-3a+WIP1i-treated and DMSO treated TP53-wt cell and they are also involved in p53 pathway? Is it correct? Which is the message of this result? Are there other p53 targets that are expressed in TP53-mut but not in TP53-wt?

Reply. As mentioned by the reviewer, the genes showed in Figure 4B are differentially expressed between  Nut-3a+WIP1i-treated and DMSO treated TP53-wt MV-4-11 cells that are also involved in the p53 pathway Hallmark signature (Figure 4A). We have added a new Supplementary Table (Table S9) summarizing all genes belonging to the signature or annotated as p53 targets (Fischer M. Oncogene 2017) that are significantly upregulated by the combined treatment (versus control) in the two cell lines. Overall, 33 genes were upregulated in the TP53-wt model and 3 genes were upregulated in the TP53-mut model, with no overlap among the analyzed cell lines.

  • In the discussion you proposed that “the combined treatment may stabilize the p53-phosphorylated form, which in turn is able to propagate the stress mediated response by activating downstream targets and triggering apoptosis”. Did you check phospho-p53 in western blot after treatment?

Reply. We have performed western blot analysis of phospho-p53 after treatment (Fig. 4C-D and Fig. S3A-B). Our data show heterogeneous results, with increased levels of phopsho-p53 in the TP53-wt MV-4-11 cell line.  We have also rephrased the sentence in the discussion section.

Minor Comments

  • In section 2.4 you affirm “the viability of 3 AML primary cells”, but in supplementary are 9 samples.

Reply. Supplementary Table 2 reports the all primary samples used for analyses.

  • Please set a correct Table 1 formatting. Variables and data sometimes are not in the same rows.

Reply. We have corrected the Table.

  • In the barplot in fig 3 and 4, sometimes DMSO bar is black and sometimes is grey. Please uniform graphs.

Reply. We have corrected the bars in the barplot.

This manuscript is a resubmission of an earlier submission. The following is a list of the peer review reports and author responses from that submission.

Round 1

Reviewer 1 Report

In this manuscript, Dr. Fontana and co-workers present data supporting the synergistic effect of MDM2 and WIP1 inhibitors in AML TP53-wildtype cells. They show that AML patients show variable PPM1D/WIP1 expression according to the cytogenetic groups and the age of the patient. They show as well that, in TP53-wildtype cells, MDMD2 and WIP1 combined inhibition increase apoptosis-mediated cell death. 

Although the results presented are interesting, the novelty of this study is reduced by the fact that this combined regimen has been proven effective in other tumour types based in the same rationale. Nevertheless, this combined treatment has not been tested in AML before, and the authors provide data on AML primary samples which provides additional interesting data. In spite of that, some of the conclusions seems a bit premature with the presented data so some major concerns need to be addressed before I could recommend its publication. 

Major Concerns:

1) Many conclusions are extracted from the lack of effect of the combined treatment on a TP53-deficient cell line NOMO-1. This supports the idea that the effect of the drug combination is mediated by TP53. Nevertheless, the behaviour of a single cell line is probably not enough to associate the effect to the TP53 deficiency as this cell lines might have many other alterations. It is necessary to check this behaviour in additional TP53-deficient cell lines.

2) As the key aspect of the paper is the response to different drugs in cell lines, proper drug-response curves with more drug concentrations and an IC50 estimation are necessary observe these responses more clearly. This is particularly important in the primary samples in where the effect seems to be subtle in the selected drug concentrations and where the differences might be clearer with higher drug concentrations.

3) Expression differences are only extracted from a single TP53-wildtype cell line (MV-4-11). Again, the results obtained in a single cell line might not be representative of the real situation in the rest of cell lines/primary samples. Some critical expression differences should be selected and tested in the whole panel of cell lines/primary samples to check the relevance of these observations.

4) It is not perfectly clear to me the relevance of the observations obtained from the GSEA analysis of the expression data (section 3.5 of the results). What is the meaning of the differences between the cell line treated with the single agents and the combined treatment? Does these differences provide any extra mechanistic information to the manuscript? If that is so, Why are they not include in the conclusions section? If this section is purely descriptive and don't provide useful information, it should be removed. Otherwise, a proper more detailed explanation of the meaning of these observations should be included in the results and conclusions sections of the manuscript. 

Minor points:

1) In Figure 4C, representative Western-blot pictures should be shown in order to check the specificity of the antibodies and how reliable are the protein quantifications.

Reviewer 2 Report

Please revise the reference style to be in accordance with the MDPI guidelines.

How did you assess normality of the distribution? Why did you use only parametric approaches, was it the case that all your data was normally distributed? Figure 1 assessed different groups. For TCGA for example you can get the FLT3 and NPM1 mutational status, so the lack of data is not an excuse in this case.   Overall the study has a nice take on MDM2 inhibitors which could add valuable information for the start of a clinical trial as MDM2 inhibitors are already used in clinical trials for AML.